# Disparities in Negation Understanding Across Languages in Vision-Language Models

## Abstract

Vision-language models (VLMs) exhibit *affirmation bias*: a systematic tendency to select positive captions ("X is present") even when the correct description contains negation ("no X"). While prior work has documented this failure mode in English and proposed solutions, negation manifests differently across languages through varying morphology, word order, and cliticization patterns - raising the question of whether these solutions serve all linguistic communities equitably. We introduce the first human-verified multilingual negation benchmark, spanning seven typologically diverse languages: English, Mandarin Chinese, Arabic, Greek, Russian, Tagalog, and Spanish. Evaluating three VLMs - CLIP, SigLIP, and MultiCLIP - we find that standard CLIP performs at or below chance on non-Latin-script languages, while MultiCLIP achieves the highest and most uniform accuracy. We also evaluate SpaceVLM, a proposed negation correction, and find that it produces substantial improvements for several languages - particularly English, Greek, Spanish, and Tagalog - while showing varied effectiveness across typologically different languages. This variation reveals that linguistic properties like morphology, script, and negation structure interact with model improvements in fairness-relevant ways. As VLMs are deployed globally, multilingual benchmarks are essential for understanding not just whether solutions work, but for whom.

## 1 Introduction

Prompts to language models are overwhelmingly given in the affirmative: "generate an image with a dog," "give me a list that contains recipes." What happens when we negate? Recent work by Alhamoud et al. (2025) demonstrates that state-of-the-art vision-language models (VLMs) frequently fail on negated queries, exhibiting *affirmation bias*: the tendency to match a caption like "there is no boat" to an image containing a boat, simply because the word "boat" appears. This failure has serious downstream consequences—in radiology, the distinction between "no pleural effusion" and "pleural effusion present" determines whether a finding is flagged or dismissed.

However, all existing evaluations of negation understanding focus exclusively on English. This is a significant gap because negation is not linguistically uniform. For example, English uses simple adverbial particles ("no," "not"), but Greek employs verbal negation with existential constructions ("$\delta\varepsilon\nu\ \upsilon\pi\acute{\alpha}\rho\chi\varepsilon\iota$" = "there does not exist"), Arabic uses cliticized markers in a right-to-left script, and Chinese relies on isolating particles with distinct semantic functions. These structural differences may influence VLM behavior in ways that English-only benchmarks cannot reflect. This is crucial for fairness, because AI language technology is disproportionately built around English and a small handful of other high-resource languages, creating a "digital language divide" that systematically disadvantages most of the world's languages (Bella et al., 2023). For example, a hospital in Athens or a research institute in Beijing may adopt VLMs without any way to evaluate how negation understanding behaves in their language. Prior work has documented systematic performance inequalities across languages in NLP more broadly (Blasi et al., 2022), and if VLMs similarly handle negation better for some linguistic communities than others, this constitutes an inequity that deserves attention—particularly as these systems enter safety-critical domains.

Our primary contribution is the first multilingual negation benchmark for VLMs, spanning seven typologically diverse languages. Using this benchmark, we find that even explicitly multilingual

Table 1: **Baseline cross-lingual negation performance.** Even MultiCLIP, the most equitable model, achieves only 41% mean accuracy, showing that multilingual training alone does not resolve negation understanding.

| Model | Mean Acc. | Std. Dev. | Max–Min Gap |
|-------|-----------|-----------|-------------|
| CLIP | 23.5% | 9.2% | 27.5% |
| SigLIP | 30.5% | 2.6% | 7.4% |
| MultiCLIP | 41.2% | 1.3% | 4.4% |

VLMs exhibit significant cross-lingual negation gaps, with performance varying by up to 27.5 percentage points across languages. We further apply SpaceVLM (Ranjbar et al., 2025)—a recently proposed negation correction—across all languages and uncover a typological pattern: the method's effectiveness correlates with how a language expresses negation morphologically. This variation is itself informative: it reveals that linguistic structure shapes model behavior in ways that inform equitable deployment.

## 2 Approach

**Benchmark construction.** We extend the English NegBench dataset (Alhamoud et al., 2025), which contains 5,914 image-caption sets from COCO (Lin et al., 2014) with 4-way multiple choice, to seven languages chosen for typological diversity: English, Spanish, Greek, and Tagalog (negation via independent particles), Russian and Arabic (morphologically complex negation), and Mandarin Chinese (isolating, with distinct negation particles for different semantic functions). These languages also span Latin and non-Latin scripts, left-to-right and right-to-left writing, and multiple language families. Translations used Google Translate followed by human verification from native speakers (30 samples/language), who checked that negation markers were correctly translated, that negated sentences remained semantically faithful to the English source, and that phrasing was natural in the target language.

**Models and evaluation.** We evaluate three contrastive VLMs representing different training regimes: CLIP (Radford et al., 2021) (primarily English-trained), SigLIP (Zhai et al., 2023) (multilingual), and MultiCLIP (Carlsson et al., 2022) (multilingual). We also evaluate after applying SpaceVLM (Ranjbar et al., 2025), which computes hybrid embeddings that decompose captions into affirmative and negated components using a threshold $\tau=0.92$ derived from English data. We measure top-1 accuracy on the 4-way caption ranking task (chance = 25%).

## 3 Results

**Even multilingual VLMs exhibit significant negation gaps.** Table 1 summarizes baseline performance. CLIP shows the most extreme disparity: English achieves 39.3% while Arabic (15.7%), Tagalog (11.8%), and Greek (18.0%) fall *below* chance, indicating systematic failure. SigLIP, trained on multilingual data, narrows this gap but still shows a 7.4-point spread. MultiCLIP achieves the most consistent performance (std. dev. 1.3%), yet its mean accuracy of 41.2% indicates that even the best multilingual model struggles with negation across the board.

**SpaceVLM as a diagnostic lens.** SpaceVLM produces substantial improvements for several languages, but the gains are not uniform—they follow a typological pattern (Table 2). Languages with negation expressed via independent particles (Miestamo, 2005), such as English, Spanish, Greek, and Tagalog, show consistent large gains across all three models (+9.0 to +27.5pp). By contrast, languages with morphologically complex negation (Miestamo, 2005) (Russian, Arabic) or distinct negation particles with different semantic functions (Miestamo, 2005) (Chinese) show smaller or variable effects.

**The pattern is robust.** We applied SpaceVLM to NegCLIP and ConCLIP—models fine-tuned specifically for negation on English data (Yuksekgonul et al., 2023) and it produces the same typo-

Table 2: SpaceVLM reveals typological patterns ($\Delta$ = SpaceVLM $-$ Baseline). Green: gain >5pp; Red: decrease >5pp. Languages with adverbial negation benefit consistently; languages with complex morphological negation show variable effects, revealing that negation structure affects behavior.

| Language | CLIP | | | SigLIP | | | MultiCLIP | | |
| | Base | Space | $\Delta$ | Base | Space | $\Delta$ | Base | Space | $\Delta$ |
|---|---|---|---|---|---|---|---|---|---|
| English | 39.3 | 62.5 | +23.3 | 34.6 | 54.6 | +20.0 | 40.6 | 66.1 | +25.5 |
| Chinese | 25.5 | 30.7 | +5.2 | 31.9 | 12.4 | −19.5 | 42.5 | 38.6 | −3.9 |
| Arabic | 15.7 | 16.6 | +0.9 | 28.3 | 30.1 | +1.8 | 40.6 | 34.5 | −6.1 |
| Greek | 18.0 | 33.9 | +15.9 | 28.1 | 46.0 | +17.9 | 38.9 | 66.4 | +27.5 |
| Russian | 20.4 | 19.1 | −1.3 | 30.4 | 19.8 | −10.6 | 43.3 | 36.3 | −7.0 |
| Tagalog | 11.8 | 50.7 | +38.9 | 27.2 | 31.0 | +3.8 | 41.6 | 50.6 | +9.0 |
| Spanish | 33.6 | 56.2 | +22.6 | 33.1 | 56.8 | +23.7 | 41.3 | 67.5 | +26.2 |

logical pattern: strong English gains (57–69%) with consistent gaps for Arabic (18–20%), Russian (27–29%), and Chinese (35–39%). This persistence across five model-solution combinations suggests it reflects a genuine property of how these languages encode negation.

## 4    DISCUSSION: IMPLICATIONS FOR EQUITABLE DEPLOYMENT

When VLMs systematically handle negation better for English speakers than, for example, for Arabic or Russian speakers, this constitutes a clear disparity in model reliability across linguistic communities. This is especially concerning in safety-critical applications: a radiology VLM that correctly interprets "no fracture" in English but fails on the Arabic or Chinese equivalent provides unequal service quality along linguistic lines. Our benchmark makes these disparities measurable. Beyond these specific results, the typological pattern we observe suggests that the *way* a language expresses negation—not just the language itself—influences model behavior. Languages where negation is adverbial and structurally similar to English benefit from corrections designed on English data; languages with morphologically distinct negation systems do not. This implies that fairness audits for multilingual AI should go beyond language coverage to consider whether the linguistic structures present in diverse languages are adequately captured by models and their corrections. Future work could explore language-family-specific calibration (e.g., tuning SpaceVLM's threshold $\tau$ per typological group). Additionally, because SpaceVLM's decomposition of captions into affirmative and negated components relies on parsing strategies suited to English syntax, the structural differences across languages suggest that fine-tuning a multilingual LLM to more accurately extract affirmative and negated components could improve performance in a multilingual context.

**Limitations.**   Our translations relied on Google Translate with human verification of 30 samples per language ($\sim$0.5% of the dataset); fully human-verified translations would strengthen the benchmark. We evaluated SpaceVLM with its default English-optimized hyperparameters, and language-specific tuning of the threshold $\tau$ may reduce the cross-lingual variation we observe. Finally, we evaluated three open-source VLMs; extending to proprietary and domain-specific models (e.g., clinical VLMs for radiology) is an important direction for future work.

## 5    CONCLUSION

We present the first multilingual benchmark for negation understanding in VLMs, revealing that even explicitly multilingual models exhibit significant cross-lingual negation gaps. By applying SpaceVLM across seven typologically diverse languages, we uncover a pattern linking negation morphology to model behavior: languages with adverbial negation benefit consistently from corrections, while morphologically complex negation systems show variable responses. This difference shows that linguistic typology influences how models interpret meaning, and that those effects have real fairness implications. As VLMs are deployed globally, benchmarks that capture this typological diversity are essential for ensuring that alignment improvements benefit all linguistic communities, not just those whose languages happen to resemble English.

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
