# OpenReview forum: "Disparities in Negation Understanding Across Languages in Vision-Language Models"
_ICLR.cc/2026/Workshop/AFAA — AFAA 2026 Poster_

### Official Review · Reviewer_3j9i · 2026-02-18
**Disparities in Negation Understanding Across Languages in Vision-Language Models**

**Rating:** 4
**Confidence:** 3

**Summary:**

The paper investigates "affirmation bias" in VLM i.e. the tendency for models to incorrectly match negated captions to images containing the negated object, on a global scale. The authors introduce a human-verified multilingual benchmark covering English, Mandarin Chinese, Arabic, Greek, Russian, Tagalog, and Spanish. They evaluate three models (CLIP, SigLIP, and MultiCLIP) and find that even multilingual training does not resolve negation gaps, with performance varying by up to 27.5 percentage points between languages.
The paper also analyses the effectiveness of SpaceVLM, a negation correction method, showing that its success is closely tied to the linguistic structure of negation in a given language.

**Strengths:**

This work addresses a gap in VLM evaluation by extending negation studies beyond English to seven typologically diverse languages.
The benchmark construction includes human verification from native speakers to ensure semantic faithfulness and natural phrasing across different scripts and language families. The research highlights serious fairness implications, particularly for safety-critical domains like radiology, where negation errors can lead to unequal service quality across linguistic communities.

The work also highlights correlation between a language's morphological expression of negation and the effectiveness of current state-of-the-art correction methods.

**Weaknesses:**

The evaluation of SpaceVLM used default English optimised hyper-parameters. While language-specific tuning of the threshold might have yielded more consistent results across languages.
While three representative open-source VLMs were tested, the study does not include proprietary or domain-specific models (e.g. clinical VLMs) that are often used in the safety critical applications mentioned.
Only about 0.5% (30 samples per language) of the translated dataset underwent human verification, this i noted by the author but limits the   validity of the work.

---

### Official Review · Reviewer_TyFn · 2026-02-21
**Interesting fairness finding on multilingual NLP, but fairness framing needs strengthening**

**Rating:** 3
**Confidence:** 4

**Summary:**

The paper introduces a multilingual negation benchmark spanning seven typologically diverse languages and evaluates three VLMs on it. The core finding is that negation correction methods developed on English transfer unevenly across languages, with improvements correlating with how a language morphologically encodes negation. The paper frames this as a fairness issue — systems that work better for English speakers than Arabic or Russian speakers constitute inequitable deployment.

**Strengths:**

This is a good fit for the workshop. The framing around "for whom do alignment improvements work" is exactly the kind of question AFAA is interested in, and the multilingual angle brings a dimension that fairness research often neglects. The typological pattern: adverbial negation languages benefit consistently, morphologically complex ones don't, is a genuinely interesting finding and is well-supported across five model-solution combinations. The benchmark itself is a real contribution; there's nothing like it currently.
The connection to safety-critical deployment (the radiology example) is well-chosen and makes the stakes concrete.

**Weaknesses:**

The fairness framing is present in the motivation, but doesn't get developed much. The paper identifies a disparity and essentially says "this is unfair" without engaging with what that means — is this a training data issue, a tokenization issue, an architectural issue? What would remediation look like? For a fairness workshop audience, that analytical gap is noticeable.
Human verification of only 30 samples per language (~0.5% of the dataset) is a real methodological concern. If negation markers are systematically mistranslated in Arabic or Russian, the results for those languages are unreliable, which is precisely where the most concerning disparities appear.
SpaceVLM is evaluated only at its default English-tuned threshold. It's hard to know how much of the "typological pattern" is genuine linguistic structure versus just a poorly calibrated hyperparameter for non-English languages.
No statistical significance testing is reported. Given how much Tagalog varies across models (+38.9pp for CLIP, +3.8pp for SigLIP), some of these differences could easily be noise.

The benchmark contribution alone is worth having in the workshop, and the typological fairness framing is novel and thought-provoking. The paper would benefit from deeper engagement with what the disparity actually means mechanistically and what fair remediation might look like — but as a workshop paper raising an important question, it does its job well.

---

### Meta-Review · Area_Chair_5Q4H · 2026-02-26

**Recommendation:** Tiny/Short Papers Track
**Confidence:** 5

**Metareview:**

The paper investigates affirmation bias in VLMs (errors on negated captions) and introduces a multilingual benchmark across seven languages, evaluating CLIP/SigLIP/MultiCLIP. While there are concerns about limited human verification (~0.5% per language), SpaceVLM being tested with English-tuned thresholds (potentially confounding cross-language conclusions), and limited coverage of proprietary/domain-specific models, reviewers value the benchmark’s novelty and the key findings. Overall, this is a meaningful workshop contribution and I recommend acceptance.

---

### Decision · Program_Chairs · 2026-03-02

Accept (Poster)